# Calibration of Low-Cost LoRaWAN-Based IoT Air Quality Monitors Using the Super Learner Ensemble: A Case Study for Accurate Particulate Matter Measurement

**DOI:** 10.3390/s25051614

**Published:** 2025-03-06

**Authors:** Gokul Balagopal, Lakitha Wijeratne, John Waczak, Prabuddha Hathurusinghe, Mazhar Iqbal, Daniel Kiv, Adam Aker, Seth Lee, Vardhan Agnihotri, Christopher Simmons, David J. Lary

**Affiliations:** Hanson Center for Space Sciences, University of Texas at Dallas, Richardson, TX 75080, USA; gokul.balagopal@utdallas.edu (G.B.); lhw150030@utdallas.edu (L.W.); john.waczak@utdallas.edu (J.W.); pxh180012@utdallas.edu (P.H.); mazhar.iqbal@utdallas.edu (M.I.); dkiv2@illinois.edu (D.K.); adam.aker@utdallas.edu (A.A.); sethlee2024@gmail.com (S.L.); 24agnihotriv@smtexas.org (V.A.); csim@utdallas.edu (C.S.)

**Keywords:** machine learning, data science, statistics, regression, calibration, optimization, IoT

## Abstract

This study calibrates an affordable, solar-powered LoRaWAN air quality monitoring prototype using the research-grade Palas Fidas Frog sensor. Motivated by the need for sustainable air quality monitoring in smart city initiatives, this work integrates low-cost, self-sustaining sensors with research-grade instruments, creating a cost-effective hybrid network that enhances both spatial coverage and measurement accuracy. To improve calibration precision, the study leverages the Super Learner machine learning technique, which optimally combines multiple models to achieve robust PM (Particulate Matter) monitoring in low-resource settings. Data was collected by co-locating the Palas sensor and LoRaWAN devices under various climatic conditions to ensure reliability. The LoRaWAN monitor measures PM concentrations alongside meteorological parameters such as temperature, pressure, and humidity. The collected data were calibrated against precise PM concentrations and particle count densities from the Palas sensor. Various regression models were evaluated, with the stacking-based Super Learner model outperforming traditional approaches, achieving an average test R^2^ value of 0.96 across all target variables, including 0.99 for PM_2.5_ and 0.91 for PM_10.0_. This study presents a novel approach by integrating Super Learner-based calibration with LoRaWAN technology, offering a scalable solution for low-cost, high-accuracy air quality monitoring. The findings demonstrate the feasibility of deploying these sensors in urban areas such as the Dallas-Fort Worth metroplex, providing a valuable tool for researchers and policymakers to address air pollution challenges effectively.

## 1. Introduction

Every year, air pollution claims more than 7 million lives, yet most regions lack affordable monitoring systems. Currently, air pollution sensors that provide research-grade observations are typically priced at least USD 15,000. As a result, there is an urgent need for sensors that are both cost-efficient and capable of providing reliable data. Over the past two decades, research has intensified to bridge this gap by improving the performance of low-cost sensors through advanced calibration techniques [1,2,3].

While expensive sensors consistently deliver accurate and reliable measurements, they are often deployed sparsely due to their high cost. In contrast, low-cost sensors, though less accurate, offer much greater spatial coverage and the potential for dense monitoring networks. The challenge lies in improving the reliability of these affordable devices. Machine learning (ML)-based calibration methods have emerged as a key solution, using advanced data processing to enhance both the sensitivity and selectivity of low-cost sensors. One approach to a smart city initiative for sustainable air quality monitoring is to integrate low-cost sensors with research-grade instruments in hybrid networks, creating a more comprehensive and cost-effective monitoring system. This integration enables both high spatial resolution and measurement accuracy, empowering researchers and policymakers to address air pollution more effectively [4].

We used machine learning to calibrate several types of sensors as part of the research conducted at the MINTS lab. This approach is motivated by the concept of using machine learning to calibrate sensors in orbiting satellites [5,6]. Several studies have explored machine learning techniques for error correction and the calibration of low-cost sensors in IoT-based air quality monitoring networks. The research by Wijeratne [7], Wijeratne et al. [8] highlights the calibration of a low-cost particulate matter (PM) sensor using a research-grade sensor, GRIMM, with a Random Forest model. Similarly, Zimmerman et al. [1] demonstrates the use of Random Forest models to calibrate low-cost air quality sensing networks, emphasizing their ability to correct sensor drift and environmental interference. Neural networks have also been applied for sensor calibration. Zhang et al. [9] discuss the use of neural networks to enhance the accuracy of low-cost light sensors. Park et al. [10] use a similar idea by employing a hybrid LSTM neural network for calibrating PM_2.5_ sensors, demonstrating improved temporal calibration by leveraging sequential dependencies in sensor data.

A key distinction among these approaches is that Random Forest models, such as those used by Wijeratne and Zimmerman, offer high interpretability and robustness in handling structured sensor data. In contrast, neural network-based methods, as explored by Zhang and Park, effectively capture nonlinear relationships and temporal trends but often require more computational resources. To leverage the strengths of different approaches, we adopt a hybrid method that combines multiple learners, aiming to enhance both accuracy and efficiency in sensor calibration.

For this, we specifically use the “Super Learner” [11], a robust ensemble method. Super Learners were selected for this study due to their unique versatility and capacity to optimally combine multiple simple and complex models, thereby minimizing the risk of overfitting. When calibrating low-cost air quality sensors, this approach proves especially valuable compared to traditional ensemble techniques like Random Forests (bagging) or XGBoost (boosting). Unlike these methods, which rely on specific aggregation mechanisms such as bagging reducing variance and boosting reducing bias, Super Learners/Stacking Models dynamically adapt to the characteristics of diverse datasets by leveraging the strengths of multiple learning algorithms through stacking. This adaptability is crucial in sensor calibration, where data variability and environmental noise pose significant challenges. By incorporating insights from various learners, Super Learners provide a more comprehensive model, making them robust against overfitting and capable of handling large, noisy datasets. This robustness ensures that low-cost sensors, often plagued by measurement uncertainty, are calibrated with high precision, making Super Learners an ideal choice for enhancing prediction accuracy across varying environmental conditions. This technique can now be utilized on a large scale to calibrate low-cost sensors deployed across the Dallas-Fort Worth metroplex. These calibrated sensors provide valuable data for both the public and policymakers, enabling precise monitoring of particulate matter (PM) concentrations at a significantly reduced cost.

For this study, we aim to calibrate affordable LoRaWAN sensors [7] against expensive research-grade sensors. LoRaWAN (Long Range Wide Area Network) [7,12,13,14] is a low-power, wide-area networking protocol specifically designed for wireless communication over extended distances. It leverages radio frequency signals to transmit data efficiently, making it particularly well suited for applications with low power requirements. In rural areas, LoRaWAN can typically achieve a range of several kilometers, while in urban environments, the range is usually from around one to two kilometers, depending on obstructions and environmental conditions. redIt’s low power requirement and long-range communication capability makes it widely applicable in various fields, including environmental monitoring, smart cities, agriculture, and industrial IoT.

These LoRaWAN air quality monitors can be deployed in remote locations where access to a mains electrical power supply is unavailable. They are equipped with solar panels, allowing them to self-charge and extend their battery life for several months or even multiple years [15]. Each sensing unit includes a GPS module, enabling the recorded data to be accurately linked to its corresponding location using the latitude and longitude coordinates. The data collected are transmitted to the LoRa gateway with relatively low error, thanks to the reliability of LoRa technology. These features make it possible to deploy sensors at a neighborhood scale without significant data loss or power interruptions. The LoRaWAN network consists of the following two key components: LoRa nodes and gateways [16]. The LoRa nodes are end devices, such as air quality monitors in our case, that collect and send data. These nodes communicate with a gateway, which serves as a bridge between the nodes and a central server or cloud system. The gateway receives radio signals from the nodes and forwards the data to the central system using backhaul connections like Ethernet or cellular networks.

Palas is the expensive research grade sensor used in this study. All the machine learning algorithms used for calibration are programmed in Python because of its coding simplicity and speed. The LoRaWAN-based sensor and the research grade reference sensors are maintained close to each other in various environmental conditions to ensure consistent and accurate calibration readings.

## 2. Materials and Methods

### 2.1. Air Quality Monitors Utilized in the Calibration Study

In this study, we utilize two types of air quality monitors. The first is the Palas Fidas Frog, a research-grade sensor valued at approximately $20,000, known for its high precision and reliability. The second is an inexpensive prototype sensor built on LoRaWAN technology, which is powered by solar energy and a battery. This prototype is designed to monitor variations in particulate matter (PM) concentrations and is equipped with a climate sensor to track weather conditions.

The detailed descriptions of the air quality monitors are as follows:

#### Research-Grade Air Quality Monitor: Palas Fidas Frog

The Palas Fidas Frog [17,18] (Figure 1) is an optical particle counter that is more affordable than Federal Equivalent Method (FEM) devices, which still provides precise measurements of particulate matter (PM). It is portable, battery-operated, and the smallest model among the Fidas aerosol monitors. The Fidas Frog measures the following parameters: PM_1.0_, PM_2.5_, PM_4.0_, PM_10.0_, Total PM Concentration, and Particle Count Density. It employs the well-established single-particle optical light scattering method for measurement.

The Fidas Frog features a detachable control panel on top, which wirelessly interfaces with an operator’s panel (such as a tablet) via a Wireless Local Area Network (WLAN). The WiFi module facilitates the transmission and reception of data using radio frequencies. The rechargeable battery unit ensures long-lasting, portable operation, making it suitable for both indoor and outdoor applications. The Fidas Frog also monitors environmental parameters such as atmospheric pressure, ambient air temperature, and relative humidity using dedicated pressure, temperature, and humidity sensors, respectively. For this study, only the aerosol (particulate matter) data from the Palas Fidas Frog was utilized.

A built-in suction pump draws ambient air into the device, where it is analyzed by the aerosol sensor. The core of the Fidas Frog is its aerosol sensor, which operates as an optical aerosol spectrometer based on the principle of Lorenz–Mie scattering [19]. Lorenz–Mie scattering occurs when light interacts with particles whose sizes are comparable to the wavelength of the light. In the measurement chamber, particles pass through a uniformly illuminated region using white light, and the scattered light impulses are detected at angles between 85° and 95°. The number of detected impulses corresponds to the particle count, while the intensity of the scattered light is proportional to the particle size. The signal detection unit processes these signals to determine particle diameters and concentrations. The sensor measures particulate matter within a size range from 0.18 μm to 93 μm. The Palas Fidas Frog records the following parameters:PM_1.0_: Particulate matter with an aerodynamic diameter ≤ 1.0 μm (unit is μg/m^3^).PM_2.5_: Particulate matter with an aerodynamic diameter ≤ 2.5 μm (unit is μg/m^3^).PM_4.0_: Particulate matter with an aerodynamic diameter ≤ 4.0 μm (unit is μg/m^3^).PM_10.0_: Particulate matter with an aerodynamic diameter ≤ 10.0 μm (unit is μg/m^3^).Total PM Concentration: The overall concentration of particulate matter across different size fractions (unit is μg/m^3^).Particle Count Density: The number of particles per unit volume of air (unit is number of particles/cm^3^ or #/cm^3^).

The measurement range for particulate matter concentrations using the Palas Fidas Frog is 0–100 mg/m^3^. For EPA-standard particulate matter concentrations, such as PM_2.5_ and PM_10.0_, the Palas measurement uncertainties are 9.7% and 7.5%, respectively, [17].

### 2.2. Low-Cost Air Quality Monitor: LoRaWAN Prototype

The cheaper commercial LoRaWAN-based sensor utilizes a comprehensive approach using an array of sensors that was necessary to collect sufficient data for machine learning calibration. The LoRa node prototype involved the use of a PM sensor and a climate sensor. The LoRa node uses a wireless communication method LoRa (long range) to transmit the collected sensor data to the central node (consists of a LoRaWAN gateway module), a system capable of connecting directly to the internet.

The LoRaWAN prototype consists of the following sensors (Figure 2):
*PPD42NS: Particle Counter*The PPD42NS (Figure 2a) [23] is the primary PM sensor in the LoRa node. It is an affordable optical sensor designed to measure parameters associated with particulate matter (PM) concentrations for two size ranges, detected via two separate channels. Channel 1 measures parameters associated with particulates larger than 1 μm in diameter, while Channel 2 measures parameters associated with particulates larger than 2.5 μm in diameter. The measurement range for PM concentration is approximately from 0 to 28,000 μg/m^3^ for both >1 μm (P1) and >2.5 μm (P2) particle sizes.The sensor operates based on the Low Pulse Occupancy (LPO) principle. When particles pass through the sensor’s optical chamber, they scatter the light emitted by an LED. This scattered light is detected by a photodiode, which outputs a pulse width-modulated (PWM) signal. The duration of the PWM signal is proportional to the particle count and size. LPO is defined as the amount of time during which this PWM signal is low during a fixed sampling interval (15 s in this study). The standard error for LPO is approximately 0.02 units for both channels under low-concentration conditions (<50 μg/m^3^). Under high-concentration conditions (≥50 μg/m^3^), the error varies nonlinearly [24,25].The following parameters are measured by the PPD42NS:
-**P1 LPO**: Represents the total time for Channel 1 (indicating the presence of particles larger than 1 μm) during which the sensor signal is low in a 15 s sampling period. It is also referred to as the *>1 μm LPO* and is measured in milliseconds.-**P1 ratio**: Represents the proportion of time the sensor signal is low for Channel 1 (indicating the presence of particles larger than 1 μm) during the sampling period. It is also referred to as the *>1 μm ratio*.-**P1 concentration**: Measures the PM concentration of particles larger than 1 μm in diameter. It is also referred to as the *>1 μm concentration* and is measured in μg/m^3^.-**P2 LPO**: Represents the total time for Channel 2 (indicating the presence of particles larger than 2.5 μm) during which the sensor signal is low in a 15 s sampling period. It is also referred to as the *>2.5 μm LPO* and is measured in milliseconds.-**P2 ratio**: Represents the proportion of time the sensor signal is low for Channel 2 (indicating the presence of particles larger than 2.5 μm) during the sampling period. It is also referred to as the *>2.5 μm ratio*.-**P2 concentration**: Measures the PM concentration of particles larger than 2.5 μm in diameter. It is also referred to as the *>2.5 μm concentration* and is measured in μg/m^3^.*BME280: Climate Sensor*The BME280 (Figure 2b) is the climate sensor used in the LoRa node. It measures air temperature (referred to as *temperature*), atmospheric pressure (referred to as *pressure*), and relative humidity (referred to as *humidity*), which are critical for understanding environmental conditions and calibrating the PM sensor. The BME280 sensor can measure temperatures ranging from −40 °C to 85 °C with an accuracy of ±0.5 °C, pressure from 300 hPa to 1100 hPa with an accuracy of ±1.0 hPa, and humidity from 0% to 100% with an accuracy of ±3% [26].

Table 1 compares the specifications of the Palas Fidas Frog and the LoRaWAN prototype.

### 2.3. Air Quality Monitoring Data Collection

Both the LoRa node and the Palas Fidas Frog were deployed outdoors at the Waterview Science and Technology Center (WSTC) on the University of Texas at Dallas (UTD) campus. To ensure uniform data collection under identical environmental conditions, the devices were placed 4 to 5 feet apart at a height of 2 m above ground level. The LoRa node recorded measurements at 15 s intervals, while the Palas sensor provided data every 30 s. The calibration site is indicated on the map (Figure 3) below with a red marker.

To prepare the data for calibration, the data from the LoRa nodes and the Fidas Frog were converted into separate CSV (Comma-Separated Values) files. The datasets consisted of 2063 data points spanning the period from 04/08/2019 to 04/10/2019.

### 2.4. Supervised ML Approach for Calibration

Over the past decade, machine learning [27] has revolutionized various fields, including science, technology, medicine, engineering, finance, and trade. It has proven particularly useful for calibrating cost-effective sensors when theoretical models are either unavailable or impractical. By analyzing relationships among multiple variables, machine learning reduces the need for manually testing numerous hypotheses. In this study, regression models [28,29,30,31] are employed for sensor calibration, as the target variables from the Palas sensor are continuous. Multivariate regression techniques are used to model the relationship between nine independent variables and six target variables. The input features include low pulse occupancy duration; low pulse occupancy ratio; particulate matter (PM) concentrations for various size fractions; and meteorological parameters such as temperature, pressure, and humidity. These features are used to develop machine learning models for each target variable, including the PM concentrations and particle count densities reported by the Palas sensor.

The most commonly used machine learning models for air quality calibration include Neural Networks, XGBoost, and Random Forests. Ref. [32] has shown that while a simple Lasso regressor performed effectively for PM_10.0_ calibration, Random Forest was more suitable for PM_2.5_ but struggled with PM_10.0_. This suggests that Random Forest performs well for smaller particulate sizes like PM_2.5_ due to its ability to capture nonlinear relationships and reduce overfitting through ensemble averaging. However, its reliance on decision tree splits makes it less effective at modeling smoother trends, as seen in PM_10.0_, where variations occur more gradually over time. Similarly, another study [33] found that XGBoost underperformed in PM_2.5_ calibration compared to feed-forward neural networks. However, neural networks require extensive hyperparameter tuning and are highly sensitive to noise, which limits their effectiveness for real-time calibration tasks [34]. Notably, a hybrid approach combining Ridge Regression and XGBoost demonstrated superior performance in PM_10.0_ calibration [35], highlighting the potential benefits of combining individual learners in a manner similar to the Super Learner approach.

In this study, we assess various supervised learning models for calibration, including Linear Regression, Ridge Regression, K-Nearest Neighbors, Neural Networks, LightGBM, Decision Trees, XGBoost, Ensemble Bagging, Random Forest, and Stacking Regression, to compare their performance and identify the most effective model.

#### Super Learner

To overcome the calibration challenges faced by common machine learning models, stacking ensembles—such as the Super Learner—offer a robust alternative [11,36,37]. Super Learner is an ensemble learning method that optimally combines multiple base models to achieve asymptotically optimal performance. It selects base learners from a predefined set and combines them using a meta-learner, which is chosen to maximize predictive accuracy. Unlike traditional single-model approaches, Super Learner stacks multiple base models and assigns optimal weights to them to minimize errors, making it a more reliable and adaptable solution for complex prediction tasks. A study by the Biostatistics Department at UC Berkeley evaluated the Super Learner on four distinct synthetic datasets, each with different underlying data patterns [38]. The results demonstrated that while the best-performing individual model varied across datasets, the Super Learner consistently performed as well as or better than the best model in each case, highlighting its adaptability. Furthermore, Super Learners have been shown to mitigate overfitting by selecting the most effective models, thereby improving generalization performance [39].

Research in behavior classification further validates the Super Learner’s effectiveness, as studies have shown that Super Learners consistently achieve higher accuracy and lower variance compared to individual models [40], reinforcing their reliability across diverse predictive tasks.

A key advantage of the Super Learner lies in its meta-learner. During training, the meta-learner minimizes estimation errors, improving both predictive performance and robustness. This approach ensures that the Super Learner often outperforms individual base models and traditional ensemble methods, such as Random Forest, XGBoost, and LightGBM, particularly in applications like air quality monitoring.

For robust and generalizable predictions, diversity among base learners is essential. Using highly similar models (e.g., multiple Random Forest regressors) can introduce bias and reduce the ensemble’s effectiveness. By incorporating a variety of models—such as Linear Regression, Decision Trees, and Neural Networks—the Super Learner effectively balances bias and variance, leading to superior generalization on unseen data. This makes it a powerful framework for enhancing accuracy and reliability in environmental applications.

### 2.5. Metric for ML Calibration

In order to evaluate the effectiveness of the machine learning algorithms used in this study, we calculated the coefficient of determination, commonly referred to as the R^2^ value [41]. The R^2^ value was computed using the following equation:$$R^{2}=1-\frac{\sum_{i=1}^{N}{\left(y_{i}-{\hat{y}}_{i}\right)}^{2}}{\sum_{i=1}^{N}{\left(y_{i}-\bar{y}\right)}^{2}}$$
where
$y_{i}$ is the actual target value for the *i*-th data point.${\hat{y}}_{i}$ is the predicted target value for the *i*-th data point.$\bar{y}$ is the mean of all target values.*N* is the total number of data points.

The R^2^ value represents the proportion of variance in the target variable that is explained by the input data. For instance, an R^2^ value of 0.9 indicates that 90% of the variance in the predicted Palas values is explained by the LoRa node sensor data, demonstrating a strong relationship between the model predictions and the observed values. The best models for each target variable are chosen based on the R^2^ value.

### 2.6. Hyperparameter Tuning for ML Models

For hyperparameter tuning, we use Random Search [42], a method in machine learning that optimizes model performance by randomly selecting hyperparameters from predefined ranges. Unlike grid search, which systematically evaluates all possible combinations of hyperparameter values from a given set, random search explores a subset of combinations by sampling values randomly. This makes random search more flexible and capable of discovering better-performing models, as grid search is restricted to a predefined grid and may overlook optimal configurations.

Random search was chosen over methods like Bayesian optimization and Tree-structured Parzen Estimation (TPE) due to its ease of implementation and lower computational complexity, as it does not require building and updating probabilistic models to guide the search. This makes it particularly advantageous for high-dimensional search spaces where simpler, computationally efficient approaches are preferred.

In this study, the performance of each model is optimized using R-squared (R^2^) as the evaluation metric. Random search is employed to identify the best hyperparameters for each model, and based on these optimal hyperparameters, the final model is trained to achieve the best possible performance.

### 2.7. Feature Ranking for ML Calibration

Once the best models for each target variable are tuned to achieve optimal results, we aim to understand the features that contribute to the variations in the target variable. To achieve this, we employ a model-agnostic approach, such as permutation importance.

Permutation importance [43] is a model-agnostic technique used to evaluate the importance of features in a predictive model. The process begins by training the model on the dataset and calculating a baseline performance metric, such as $R^{2}$. To measure the importance of a specific feature, its values are shuffled, thereby breaking the relationship between the feature and the target variable. The model is then re-evaluated, and the change in the performance metric is observed. A significant drop in performance indicates that the feature plays an important role in the model.

This procedure is repeated for each feature, and the features are ranked based on the extent of the performance drop. Features causing the largest decrease in performance are considered the most important. Permutation importance provides an intuitive and straightforward method for interpreting model predictions and understanding the influence of individual features on the target variable.

### 2.8. Basic ML Calibration Work Flow

Figure 4 shows the basic workflow for this study.

### 2.9. Preprocessing of LoRa and Palas Sensor Data

The LoRa and Palas data CSV files were initially combined into a single dataset, where the combined data were sampled at 30 s. The combined dataset contained 15 columns as follows: 9 input features from the LoRa node and 6 target variables from the Palas sensor, which served as the model outputs.

The data were cleaned and partitioned into training and testing sets to train machine learning models using supervised regression algorithms. The train test split used was 80:20. Each model was trained to predict one of the target variables based solely on the input features from the LoRa node.

## 3. Results

Multivariate nonlinear nonparametric machine learning regression models were developed for each target variable. All models, except the stacking models, were employed for this part of the analysis.

The performance for both training and testing was assessed based on the R^2^ value achieved for each target variable. The average R^2^ value was used to identify the best-performing model across all target variables in both training and testing scenarios. Table 2 provides a comprehensive comparison of models based on their average test R^2^ values across multiple target variables, including PM_1.0_, PM_2.5_, PM_4.0_, PM_10.0_, Total PM Concentration, redand particle Count density.

Table 2 also includes the average R^2^ values as well as individual target variable R^2^ values, calculated separately for training and test datasets, offering a holistic view of model performance. During the testing phase, the Random Forest and Ensemble Bagging models performed the best, with an average R^2^ value of 0.93 across all variables.The bagging models performed well across various target variables and it may be due to its robustness against noise. These models were followed by the Light Gradient Boosting Machine and Extreme Gradient Boosting models. For the training data, the Decision Tree and Extreme Gradient Boosting models emerged as the top performers, achieving an average R^2^ value of 1.00 across all target variables. These were followed by the Random Forest and Ensemble Bagging models, which also demonstrated strong performance.

When comparing training and testing results, it is evident that tree-based models consistently performed well, achieving R^2^ values ranging from 0.99 to 1 during training. However, in the testing phase, while the overall average R^2^ values remained high (above 0.90), some individual target variables experienced a drop in R^2^ values below 0.90. This discrepancy between training and testing performance suggests a possibility of overfitting, particularly for models that achieved perfect R^2^ values of 1 during training but showed reduced performance during testing.

To address this potential overfitting, Super Learner models were implemented to improve generalization and ensure robust performance across all target variables. Stacking learners combine simpler and more complex models to achieve improved test performance. Various combinations of models were explored to identify the best-performing learners. The table below illustrates this.

Table 3 presents the best stacking model combinations for each target variable, evaluated based on training and test R^2^ values across all target variables. While hyperparameter tuning was performed using Random Search, many of the best-performing models were obtained with default parameters.

The Random Forest emerged as the most frequently used meta learner, achieving near-perfect training R^2^ values of 1.00 for multiple target variables, including PM_1.0_, PM_2.5_, PM_4.0_, and Particle Count Density, with test R^2^ values remaining high at 0.99, indicating strong generalization. This may be attributed to Random Forest’s robustness against noise and its ability to handle complex, nonlinear relationships effectively. Simpler models, such as K-Nearest Neighbors (KNN), consistently appeared as base learners across all stacking combinations, particularly for PM variables, highlighting their critical role in capturing local data patterns.

For PM_10.0_ and Total PM Concentration, slightly lower test R^2^ values (0.91 and 0.86, respectively) were observed despite high training R^2^ values, suggesting higher variance in the test data for these variables. Neural networks, used as meta learners for these targets, likely helped address the nonlinearity in the data. Linear models, including Linear Regression and Ridge Regression, also proved effective as base learners, particularly for Particle Count Density, which achieved a near-perfect R^2^ value of 0.99 in testing.

The close alignment between training and test R^2^ values across most target variables indicates minimal overfitting, even for variables like PM_10.0_ and Total PM, where test performance was relatively lower but remained robust. The average train R^2^ value for stacking models with various base learner and meta-learner combinations across all target variables was 0.99, while the average test R^2^ value for the same combinations was 0.96. This highlights the effectiveness of the stacking framework in calibrating cost-effective sensors used in our study.

The scatter plots, quantile–quantile plots, permutation importance ranking plots, and the error distribution plots corresponding to the best-performing stacking model for each target variable are presented below (Figure 5, Figure 6, Figure 7 and Figure 8).

The scatter plots in Figure 5 compare the actual values, measured by the reference sensor, and the predicted values generated by the hyperparameter-optimized stacking models for PM_1.0_, PM_2.5_, PM_4.0_, PM_10.0_, Total PM Concentration, and Particle Count Density. The x-axis represents the actual measurements obtained from the reference sensor (Palas), while the y-axis represents the predictions generated by the stacking regression models using inexpensive LoRa sensor data as input features. Each plot includes a 1:1 reference line to indicate perfect agreement between predicted and actual values. Marginal probability density functions (PDFs) are shown above (for actual values) and to the right (for predicted values), providing a visual summary of the distribution of training (blue) and testing (orange) data. The train-test split and the coefficient of determination (R^2^) values, which quantify the model’s predictive accuracy, are included in the legend.

The R^2^ values for the testing set are exceptionally high (0.99) for PM_1.0_, PM_2.5_, PM_4.0_, and Particle Count Density, indicating near-perfect predictive performance for these variables. These results highlight the stacking model’s ability to effectively capture the variability in smaller particulate matter s, which typically remain suspended in the air for longer durations. For PM_10.0_ and Total PM Concentration, the testing R^2^ values are slightly lower, at 0.91 and 0.86, respectively.

When comparing the Super Learner model with the Random Forest Model, the results indicate notable improvements across most particulate matter fractions. For instance, the Super Learner improved the test R^2^ for PM_2.5_ from 0.98 (Random Forest) to 0.99, and for PM_1.0_ and PM_4.0_, while the R^2^ values increased from 0.97 to 0.99. For PM_10.0_, the improvement was more modest, increasing from 0.89 to 0.91, while Total PM Concentration saw an increase from 0.80 to 0.86.

The lower R^2^ values for PM_10.0_ and Total PM Concentration can be attributed to the increased variability associated with larger particle size fractions, which tend to sediment more quickly and are more influenced by localized emission sources. This may also indicate that the primary sources of particulate pollution predominantly emit smaller particles, contributing to the relatively higher predictive accuracy for smaller size fractions.

Notable deviations from the 1:1 line occur at higher concentration ranges, especially for PM_10.0_ and Total PM Concentration. These deviations suggest the need for further model refinement to improve predictions for extreme values (indicated by the higher error in the PM_10.0_ and Total PM Concentration predictions, as shown in Figure 8). However, the marginal Probability Density Functions (PDFs) confirm that the overall predicted distributions align closely with the actual distributions, underscoring the robustness of the stacking models for most scenarios. These results suggest that the stacking framework is highly effective for calibrating cheaper sensors to predict particulate matter concentrations and particle count density.

Figure 6 shows quantile–quantile (Q-Q) plots comparing the actual quantiles of the target variables with the predicted quantiles generated by the stacking regression models. The x-axis represents the actual quantiles derived from the distribution of the target variable measured by the reference sensor (Palas, using the test dataset), while the y-axis represents the predicted quantiles derived from the stacking model predictions based on the test dataset of the reduced-cost sensor node (LoRa prototype). The gray dashed line represents the ideal response, where the actual and predicted quantiles perfectly align.

The pink, orange, green, red, and purple diamonds correspond to the 0 th, 25th, 50th, 75th, and 100th percentiles, respectively. For all target variables, PM_1.0_, PM_2.5_, PM_4.0_, PM_10.0_, Total PM Concentration, and Particle Count Density, the data closely follow the ideal line, particularly between the 25th and 75th percentiles. The Q-Q plots indicate that the predicted distributions align well with the actual distributions, as evidenced by the near-linear behavior across the central quantiles (25th to 75th). This strong alignment across percentiles demonstrates that the stacking regression models are effective for calibrating cheaper sensors to accurately predict particulate matter concentrations and particle count density.

Figure 7 displays the permutation importance rankings for each of the target variables, including PM_1.0_, PM_2.5_, PM_4.0_, PM_10.0_, Total PM Concentration, and Particle Count Density. In these plots, the input features are ordered in decreasing order of importance, with the most significant variables for calibration listed at the top and the least significant at the bottom.

For the target variables the input features—humidity, pressure, and temperature—consistently emerge as the top three features contributing to accurate calibration. Specifically, humidity ranks as the most influential parameter for PM_1.0_, PM_2.5_, PM_4.0_, and Particle Count Density. This may be attributed to the hygroscopic growth of particles, where water vapor condenses onto the particles, causing them to increase in size [44,45,46,47].

Pressure and temperature also contribute significantly to the prediction of all target variables. For PM_10.0_, temperature emerges as the most important feature. This may be because an increase in temperature may lead to higher PM_10.0_ concentrations due to phenomena such as wildfires and dust storms, which are significant sources of PM_10.0_ [48,49,50]. On the other hand, pressure is the most important feature contributing to Total PM Concentration. This may be due to a positive correlation between pressure and PM Concentration, as higher atmospheric pressure tends to trap pollutants closer to the surface, thereby increasing PM Concentration [51,52,53,54].

Additionally, features such as particle size LPO (>1 μm LPO, >2.5 μm LPO), particle size ratios (>1 μm ratio, >2.5 μm ratio), and particle size concentrations (>1 μm Concentration, >2.5 μm Concentration) exhibit comparatively lower importance as the target variable transitions from smaller particulate fractions, such as PM_1.0_, to larger fractions, such as PM_10.0_ and Total PM Concentration.

The consistent dominance of meteorological factors across all target variables underscores the necessity of including humidity, pressure, and temperature measurements in effectively calibrating the low-price LoRaWAN air quality monitor prototype. These factors are essential for achieving accurate predictions and reducing errors associated with sensor limitations.

Figure 8 illustrates the error distribution for each target variable, including PM_1.0_, PM_2.5_, PM_4.0_, PM_10.0_, Total PM Concentration, and Particle Count Density. These plots show the frequency of prediction errors in the test data, calculated as the difference between actual test data and predicted test data for each target variable. An arbitrary error threshold of ±5 units was used in this analysis.

From the plots, it can be observed that for target variables such as PM_1.0_, PM_2.5_, and PM_4.0_, the error distributions are tightly centered around zero, with most of the errors ranging between −1 and 1. In contrast, the error distributions for PM_10.0_, Total PM Concentration, and Particle Count Density show larger deviations. For PM_10.0_, errors extend from approximately −4 to +8, while Total PM Concentration errors range from −15 to +20.

Interestingly, the error distribution for Particle Count Density exhibits a wider spread, with errors ranging from −40 to +20, despite a high R^2^ value indicating strong correlation. This discrepancy may be attributed to the wide range of Particle Count Density values, where even small relative prediction errors can translate into large absolute errors.

The integration of low-cost air quality sensors with advanced machine learning-based calibration models is transforming air quality monitoring in smart cities. These calibrated sensors offer a reliable and cost-effective alternative to conventional dense monitoring networks, making large-scale air pollution tracking more accessible.

One notable example is Denver’s “Love My Air” program, where the City of Denver’s Department of Public Health and Environment [55] has deployed calibrated low-cost sensors throughout the city, particularly in schools. The program provides real-time air quality data through user-friendly dashboards, enabling students, staff, and community members to monitor local air conditions. Similarly, the New York Community Air Survey (NYCCAS) [56] employs a network of low-cost sensors to measure air quality variations across New York City, with a particular focus on low-income neighborhoods and environmental justice sites. The Breathe London [57] project has implemented a comparable strategy, placing air quality monitors in schools, hospitals, and construction sites to enhance city-wide coverage and inform decision-making. In Hawaii, solar-powered low-cost air quality monitors have been deployed to inform residents of local air quality conditions using wireless communication systems [58]. These monitors provide continuous updates, even in remote areas, making real-time air quality data accessible to local communities.

## 4. Challenges and Future Work

The calibration process faces several key challenges, primarily due to the sensitivity of sensor data to environmental variability and noise. Sudden changes in environmental factors—such as temperature spikes during wildfires, abrupt shifts in wind direction, and variations in sampling locations—can introduce inconsistencies and affect sensor accuracy. For instance, temperature fluctuations during wildfires or rapidly shifting wind patterns can cause deviations in sensor readings, complicating the calibration process [59].

Another significant limitation is the restricted dataset, consisting of only 2063 data points collected over a three-day period. To improve robustness, future data collection should extend over several months to capture seasonal and extreme environmental conditions. This would ensure that both the research-grade sensors and low-cost prototypes are exposed to diverse scenarios, including heatwaves, rainfall, and snowfall. Without such extended data, the calibration’s reliability under extreme conditions cannot be fully evaluated. Additionally, the cheaper optical sensors used in the prototype are prone to failure under adverse weather conditions, such as condensation-induced drift during heavy rainfall or snowfall [60]. Future improvements will address these challenges by incorporating additional durable PM sensors, and sensors measuring nitrogen dioxide (NO_2_) and carbon dioxide (CO_2_), to enhance the system’s applicability and resilience across various environmental settings [4].

The implementation of the Super Learner model introduces a computational tradeoff compared to simpler models [61]. Its structure requires training multiple base learners and a meta-learner, resulting in higher computational costs. However, the Super Learner consistently outperforms simpler models in prediction accuracy, making the additional expense worthwhile, particularly as the dataset size scales up. For larger datasets, its added complexity often leads to significantly improved performance, despite potential scalability concerns [11].

One major contributor to the computational overhead is the hyperparameter tuning of both base and meta-learners [39]. This issue can be mitigated through parallelization, where individual base learners are trained simultaneously to improve efficiency [62]. Another limitation of the Super Learner is its reduced interpretability compared to simpler models, which may be an important consideration for certain applications. Ultimately, the decision to use a Super Learner requires balancing computational complexity with the benefits of enhanced predictive performance.

## 5. Conclusions

This study highlights the successful calibration of low-cost LoRaWAN-based IoT air quality monitors using research-grade sensors, such as the Palas Fidas Frog, through machine learning techniques. The Stacking Regressor demonstrated near-research-grade accuracy in predicting PM concentrations across various size fractions, outperforming traditional tree-based models. These findings demonstrate the feasibility of deploying affordable sensors, enhanced by advanced models like Super Learners, to achieve reliable air quality monitoring in under-resourced regions.

A key outcome of this study is the identification of meteorological variables—temperature, pressure, and humidity—as critical contributors to accurate calibration. The feature importance analysis underscores the necessity of integrating comprehensive environmental data to improve the precision and reliability of low-cost air quality monitors. By incorporating these factors, the calibration process ensures better measurement accuracy even in challenging, dynamic environmental conditions.

When properly calibrated and integrated with machine learning techniques, low-cost sensors offer a promising, scalable solution to comprehensive air quality management. They reduce the financial barriers typically associated with traditional sensor networks while significantly enhancing the spatial resolution of air quality data. This improved resolution allows for more targeted and effective pollution mitigation strategies, making air quality monitoring efforts more impactful and accessible.

The improvement in pollutant measurement accuracy directly supports global sustainability objectives, particularly the United Nations Sustainable Development Goals (SDGs) [63]. By enabling the deployment of dense monitoring networks, especially in resource-constrained regions, this approach addresses critical gaps in air quality data collection and contributes to SDG 3 [64] (Good Health and Well-Being) by helping reduce deaths and illnesses caused by air pollution [65].

Furthermore, these efforts align with SDG 7 [66] (Affordable and Clean Energy) by fostering sustainable environmental monitoring practices and SDG 11 [67] (Sustainable Cities and Communities) by minimizing the environmental impact of urban areas and promoting sustainable urban development. The deployment of low-cost, calibrated air quality networks also supports SDG 9 [68] (Industry, Innovation, and Infrastructure) by encouraging the adoption of clean, innovative technologies in smart city infrastructure, paving the way for environmentally sustainable urban growth.

In summary, this study demonstrates how advancements in low-cost sensor calibration, driven by machine learning, can lead to the development of cost-effective and scalable air quality monitoring systems. These systems are critical for addressing the global air pollution crisis, offering impactful solutions for public health, environmental sustainability, and the transition toward smarter, cleaner cities.

## Figures and Tables

**Figure 1 sensors-25-01614-f001:**
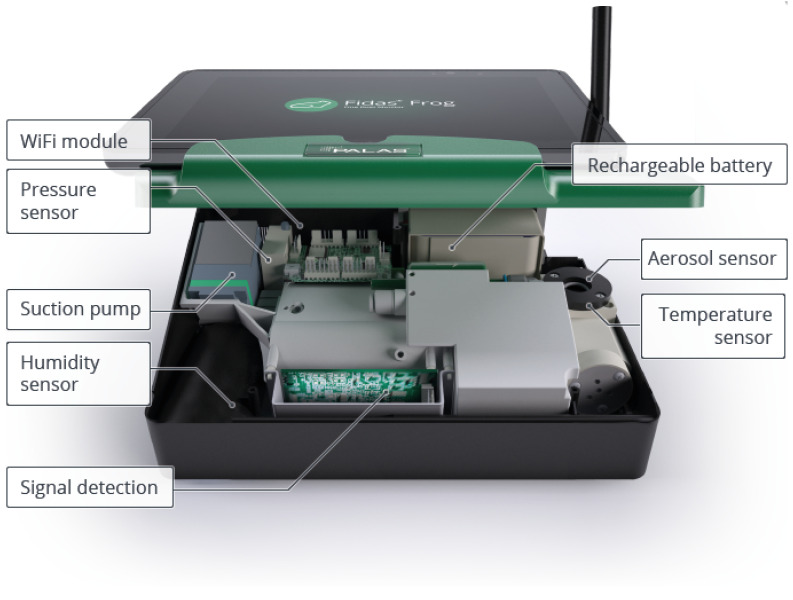
Annotated figure of Palas Fidas Frog from the Palas product information page [18].

**Figure 2 sensors-25-01614-f002:**
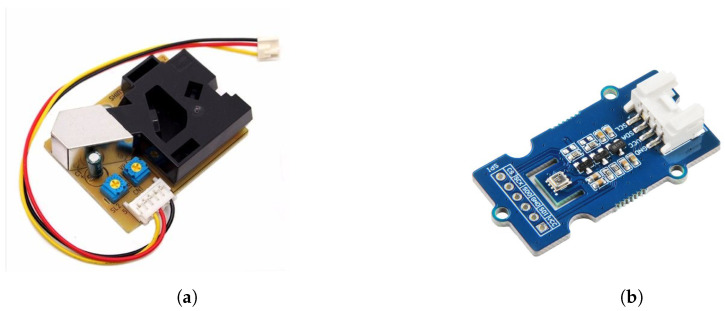
Sensors used in the low-cost LoRaWAN air quality monitor. (**a**) PPD42NS—The PM sensor used in the LoRaWAN-based air quality monitor which measures particulate matter with sizes larger than 1 μm and 2.5 μm [20]. (**b**) BME280—The climate sensor used in the LoRaWAN-based air quality monitor which measures air temperature, atmospheric pressure, and relative humidity [21,22].

**Figure 3 sensors-25-01614-f003:**
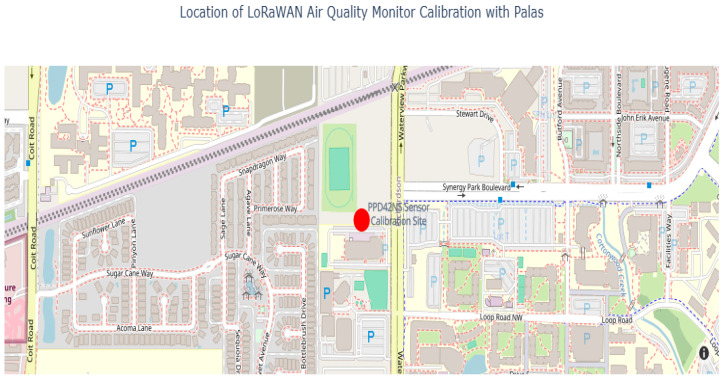
Calibration site for LoRaWAN air quality monitor deployment (indicated by the red marker and labeled in blue).

**Figure 4 sensors-25-01614-f004:**
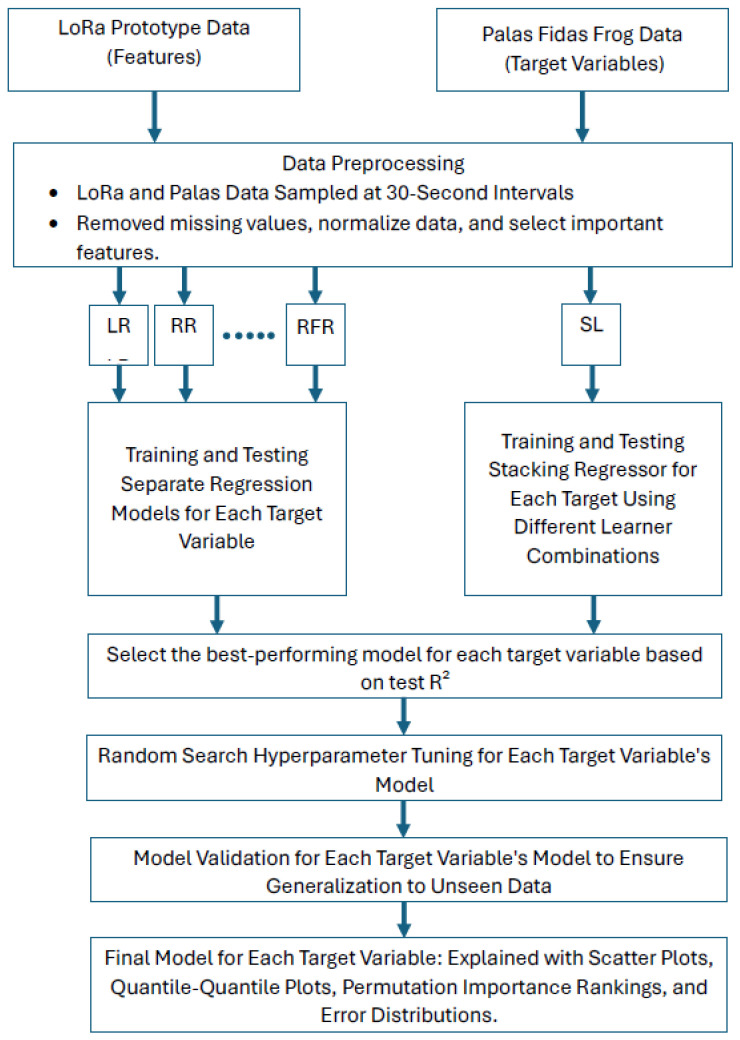
Overview of the calibration workflow using machine learning models. The process starts with LoRa prototype sensor data (features) and Palas Fidas Frog sensor data (target variables). Data preprocessing includes resampling at 30-s intervals, handling missing values, normalizing data, and selecting relevant features. Various regression models are trained separately for each target variable, along with a Stacking Regressor (SL) that combines different learner combinations. The best-performing model for each target variable is determined based on test R^2^, followed by hyperparameter tuning using random search. The final models undergo validation to ensure generalization to unseen data and are evaluated through scatter plots, quantile–quantile plots, permutation importance rankings, and error distributions.

**Figure 5 sensors-25-01614-f005:**
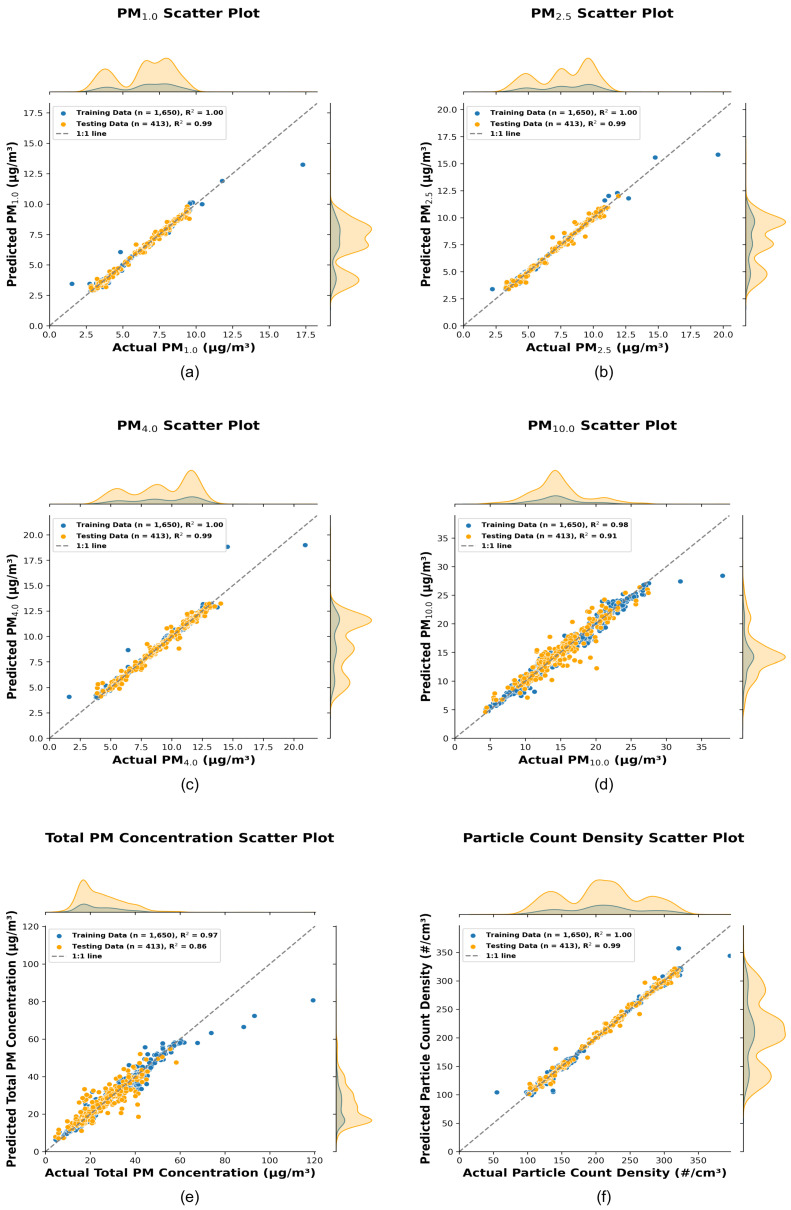
Scatter plots (**a**–**f**) illustrate the performance of hyperparameter-optimized stacking models for PM_1.0_, PM_2.5_, PM_4.0_, PM_10_, Total PM Concentration, and Particle Count Density, respectively. The blue and orange dots represent the training and testing datasets. Marginal distributions of the actual data (**top**) and predicted data (**right**) provide additional insights into the model’s performance. The legends include the train-test split count and R^2^ values, quantifying the accuracy and overall effectiveness of the predictions.

**Figure 6 sensors-25-01614-f006:**
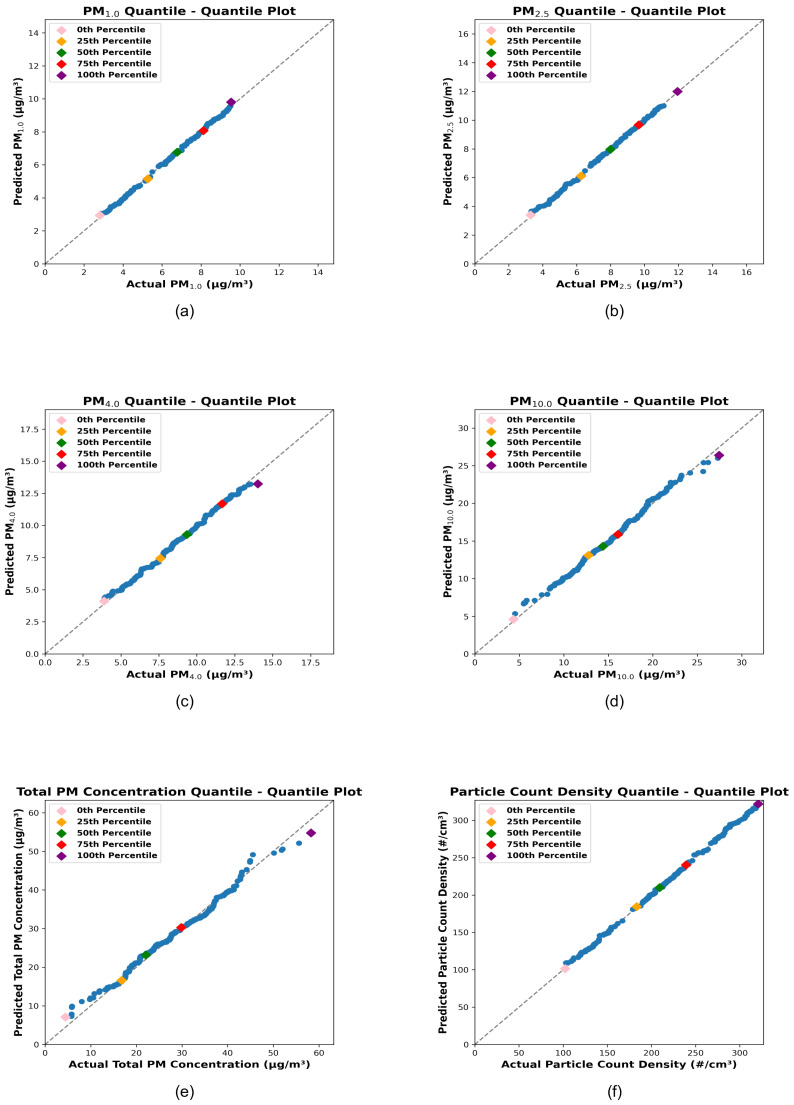
The plots (**a**–**f**) illustrate the Quantile–Quantile (QQ) plots for the hyperparameter-optimized stacking models for PM_1.0_, PM_2.5_, PM_4.0_, PM_10_, Total PM Concentration, and Particle Count Density, respectively. The quantiles of the actual test data are represented on the x-axis, while the quantiles of the predicted test data are shown on the y-axis. The 0th, 25th, 50th, 75th, and 100th quantiles are marked as pink, orange, green, red, and purple diamonds, respectively. These plots provide a visual comparison of the distribution alignment between the actual and predicted test data, demonstrating the performance of the stacking models.

**Figure 7 sensors-25-01614-f007:**
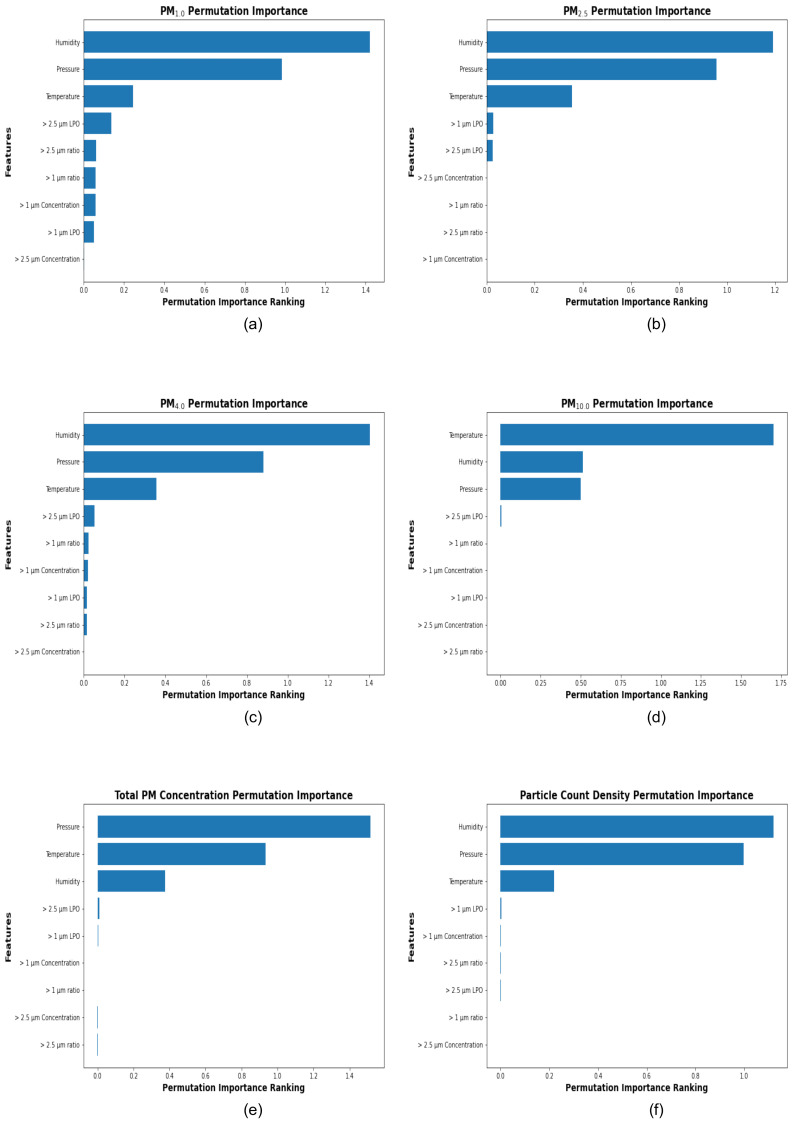
The plots (**a**–**f**) illustrate the feature importance rankings of the hyperparameter-optimized stacking models for PM_1.0_, PM_2.5_, PM_4.0_, PM_10_, Total PM Concentration, and Particle Count Density, respectively. The permutation importance rankings are displayed as horizontal bar charts, with the most important feature ranked at the top, followed by other features in descending order of importance. These rankings highlight the relative contribution of each feature to the prediction accuracy of the stacking models.

**Figure 8 sensors-25-01614-f008:**
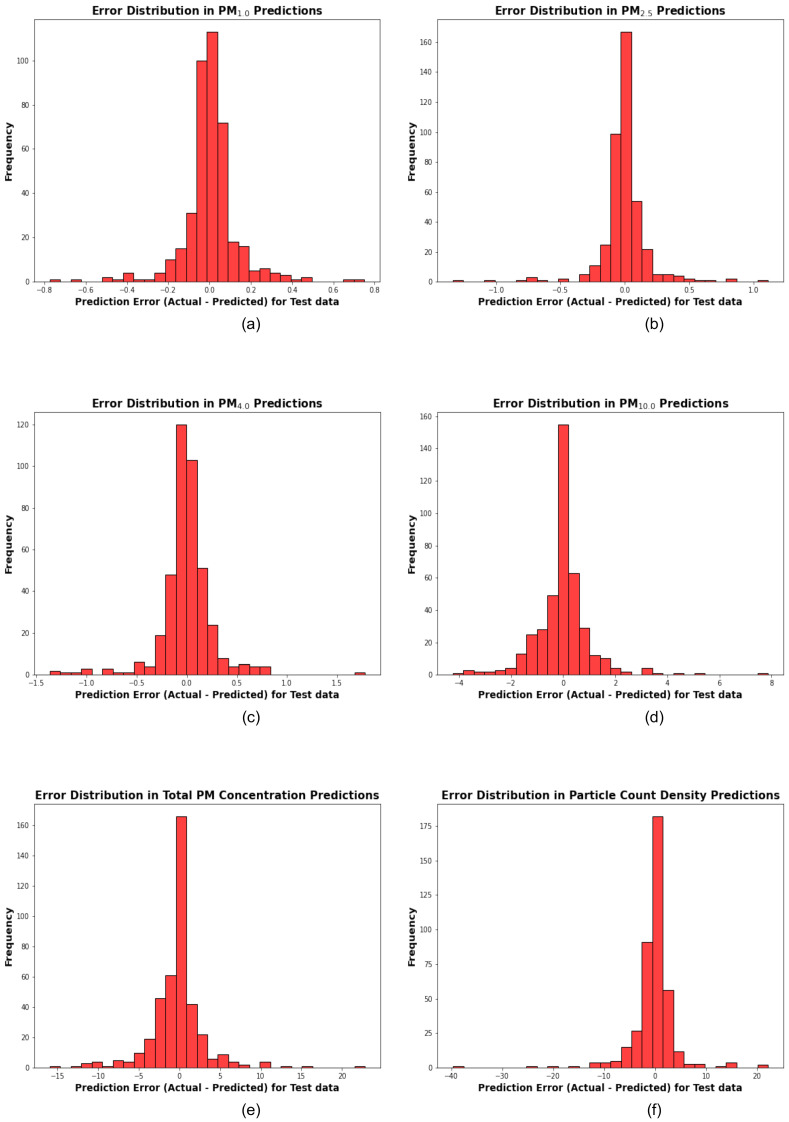
The plots (**a**–**f**) illustrate the error distribution of the hyperparameter-optimized stacking models for PM_1.0_, PM_2.5_, PM_4.0_, PM_10_, Total PM Concentration, and Particle Count Density, respectively. The error distributions are displayed as histograms (in red color). The y-axis represents the frequency of errors, while the x-axis shows the prediction error for test data, calculated as the actual test data–predicted test data. The threshold for identifying significant errors is set at ±5 for each target variable.

**Table 1 sensors-25-01614-t001:** Comparison of Palas Fidas Frog vs. LoRaWAN prototype.

Specification	Palas Fidas Frog	LoRaWAN Prototype
**Measurement Range (PM)**	0–100 mg/m^3^	0–28,000 μg/m^3^
**Particle Size Range**	0.18–93 μm	>1 μm, >2.5 μm
**Measurement Uncertainty**	9.7% (PM_2.5_), 7.5% (PM_10_)	2%
**Power Source**	Battery-operated	Solar-powered
**Cost**	Approximately $20,000 USD	$100–$200 USD

**Table 2 sensors-25-01614-t002:** Comparison of training and testing R^2^ metrics across models and target variables.

Model	PM_1.0_		PM_2.5_		PM_4.0_		PM_10.0_		Total PM Conc.		Particle Count Density		Avg. R^2^	
	**Train**	**Test**	**Train**	**Test**	**Train**	**Test**	**Train**	**Test**	**Train**	**Test**	**Train**	**Test**	**Train**	**Test**
Random Forest	1.00	0.97	1.00	0.98	1.00	0.97	0.99	0.89	0.97	0.80	1.00	0.99	0.99	0.93
Ensemble Bagging	1.00	0.96	1.00	0.97	1.00	0.97	0.98	0.90	0.96	0.80	1.00	0.99	0.99	0.93
Light Gradient Boosting Machine	1.00	0.95	1.00	0.95	0.99	0.96	0.96	0.90	0.92	0.80	1.00	0.98	0.98	0.92
Extreme Gradient Boosting	1.00	0.97	1.00	0.98	1.00	0.96	0.99	0.87	0.99	0.71	1.00	0.99	1.00	0.91
Decision Tree	1.00	0.97	1.00	0.98	1.00	0.96	1.00	0.81	1.00	0.67	1.00	0.99	1.00	0.90
Neural Network	0.95	0.88	0.94	0.88	0.93	0.88	0.63	0.59	0.45	0.40	0.95	0.92	0.81	0.76
K-Nearest Neighbors	0.97	0.86	0.96	0.86	0.96	0.87	0.82	0.59	0.73	0.46	0.96	0.90	0.90	0.76
Linear Regression	0.48	0.46	0.51	0.50	0.56	0.55	0.16	0.17	0.21	0.21	0.40	0.38	0.39	0.38
Ridge Regression	0.48	0.46	0.51	0.50	0.56	0.55	0.15	0.17	0.21	0.21	0.40	0.38	0.39	0.38

**Table 3 sensors-25-01614-t003:** Stacking models with the best-performing base and meta learners for both training and testing, evaluated using R^2^ metrics for all target variables.

Target Variable	Base Learners	Meta Learner	R^2^ Train	R^2^ Test
PM_1.0_	Linear Regression, K-Nearest Neighbors, Extreme Gradient Boosting, Neural Network	Random Forest	1.00	0.99
PM_2.5_	K-Nearest Neighbors, Decision Tree, Bagging Regressor, Neural Network	Random Forest	1.00	0.99
PM_4.0_	Linear Regression, K-Nearest Neighbors, Decision Tree, Extreme Gradient Boosting	Random Forest	1.00	0.99
PM_10.0_	K-Nearest Neighbors, Random Forest, Extreme Gradient Boosting, Light Gradient Boosting Machine	Neural Network	0.98	0.91
Total PM Concentration	K-Nearest Neighbors, Random Forest, Extreme Gradient Boosting, Light Gradient Boosting Machine	Neural Network	0.97	0.86
Particle Count Density	Linear Regression, Ridge Regression, Decision Tree, Light Gradient Boosting Machine	Random Forest	1.00	0.99

## Data Availability

The data used in this study can be accessed using the following link: https://zenodo.org/records/14776153 (accessed on 10 January 2025). The code for calibrating the LoRa node prototype can be found at the following link: https://github.com/gokulbalagopal/Calibration-of-LoRaNodes-using-Super-Learners (accessed on 10 January 2025).

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
