# Peer review of "Calibration of Low-Cost LoRaWAN-Based IoT Air Quality Monitors Using the Super Learner Ensemble: A Case Study for Accurate Particulate Matter Measurement"

_sensors, 2025, doi:10.3390/s25051614_

Round 1
Reviewer 1 Report
Comments and Suggestions for Authors
Thanks to the authors for an interesting manuscript. I read the manuscript with pleasure.
For me little surprising is the relative age of the measured data - 2019. That's 5 years ago! This fact does not affect the quality of the manuscript.
In general, the research described in the manuscript is reliable and relies on the authors' good understanding of working principles with Particulate matter sensors. The PPD42NS: Particle matter sensor does not specified as working at sub-zero temperatures according to the Data Sheet. The accuracy of the measurements is that in Dallas (Texas) there are no negative temperatures (frozen) from August to October. Here is a high correctness of experimental data, but the question remains about the application of developed mesured system in other climatic conditions different from Texas. This question can only be answered by an experiment and the correctness of the applied machine learning based on the obtained data (a new experiment is needed - but that are new research).
If the Authors supplement the manuscript with a photograph (external view) and, for example, a mark of the location on the Google-map of the experimental PM device that collected the data during experiment, it would be good. Because, so many factors depend on the location of the monitoring system relative to artificial factors generating particles (highways, industrial enterprises, etc.) which have influence on Super Learner Ensemble model (That are discussed by authors in manuscript).
Author Response
Dear Reviewer,
I sincerely appreciate your time and effort in reviewing my manuscript. Your insightful comments and constructive feedback have been invaluable in strengthening the clarity and depth of my study. I am grateful for your positive remarks regarding the research's reliability and its methodological approach.
In response to your comments, I have carefully addressed your concerns and incorporated the necessary revisions to enhance the manuscript. Specifically, I have:
- Acknowledged the cheaper air quality monitor PM sensor’s limitations in sub-zero temperatures and noted that the design will be updated to include a sensor suitable for colder climates.
- Added a marked location on Google Maps to provide better context regarding the calibration site.
I appreciate your thoughtful suggestions, which have contributed to improving the manuscript. I shared my responses to your comments in a separate document. I look forward to your further insights and any additional feedback you may have.
Thank you once again for your time and valuable input.
Best regards,
Gokul

Reviewer 2 Report
Comments and Suggestions for Authors
This study calibrates an affordable, solar-powered LoRaWAN air quality monitoring prototype using the research-grade Palas Fidas Frog sensor. Some issues should be addressed as follows.
1. The motivations, novelty and main contributions of this paper should be emphasized in the abstract.
2. As the background of this paper includes IoT and machine learning, recent related works should be introduced, such as Distributed DDPG-based resource allocation for age of information minimization in mobile wireless-powered Internet of Things, IEEE IoTJ.
3. More related works about IoT, machine learning, and sensor networks should be introduced and compared.
4. Why do authors consider to adopt Palas Fidas Frog [14] in figure 1?
5. The source of figure 2 should be provided.
6. Since there are many existing machine learning approaches, their characteristics should be introduced with evidence and details. Besides, what’s the main idea of Super Learner [28,29]? [28] and [29] are published twenty years ago.
7. Authors need to clearly show the strength of the super learner with introduction of related works.
8. Why do authors adopt random search for hyperparameter tuning?
9. The relationship between subsections is hard to see. Authors need to improve the logic and structure of this paper.
10. More details and explanations should be provided for figure 3. Besides, the clarity and readability of figure 3 should be improved.
Author Response
Dear Reviewer,
I sincerely appreciate the time and effort you have taken to provide detailed and constructive feedback on my manuscript. Your insights have been invaluable in refining my work, and I have carefully addressed each of your comments to enhance the clarity, structure, and scientific rigor of the paper.
In the revised manuscript, I have incorporated the suggested improvements, including a clearer emphasis on the study’s motivations and contributions, expanded discussions on related works, and refinements to the methodology and figures. I have also clarified my rationale for selecting specific approaches, such as random search for hyperparameter tuning. Additionally, I have improved the logical flow of the manuscript by restructuring certain sections and enhancing the readability of key figures.
A detailed response to each of your comments was submitted through the mdpi portal. To ensure transparency, all revisions are highlighted in red within the manuscript.
Once again, I truly appreciate your valuable feedback, which has helped strengthen the quality of this work. Please let me know if any further modifications are needed.
Best regards,
Gokul Balagopal

Round 2
Reviewer 2 Report
Comments and Suggestions for Authors
Authors have adequately addressed my previous comments.